# Adrenomedullin: A Novel Therapeutic for the Treatment of Inflammatory Bowel Disease

**DOI:** 10.3390/biomedicines9081068

**Published:** 2021-08-23

**Authors:** Shinya Ashizuka, Toshihiro Kita, Haruhiko Inatsu, Kazuo Kitamura

**Affiliations:** 1Division of Gastroenterology and Hepatology, Department of Internal Medicine, Faculty of Medicine, University of Miyazaki, Miyazaki 889-1692, Japan; inacchi@med.miyazaki-u.ac.jp; 2Department of Projects Research, Frontier Science Research Center, University of Miyazaki, Miyazaki 889-1692, Japan; toshihiro_kita@med.miyazaki-u.ac.jp (T.K.); kazuokit@med.miyazaki-u.ac.jp (K.K.)

**Keywords:** adrenomedullin, inflammatory bowel disease, ulcerative colitis, Crohn’s disease, translational research, drug discovery research

## Abstract

Adrenomedullin (AM) is a bioactive peptide with various physiological functions, including vasodilation, angiogenesis, anti-inflammation, organ protection, and tissue repair. AM suppresses inflammatory cytokine production in the intestinal mucosa, improves vascular and lymphatic regeneration and function, mucosal epithelial repair, and immune function in the intestinal bacteria of animal models with intestinal inflammation. We have been promoting translational research to develop novel therapeutic agents for inflammatory bowel disease (IBD) using AM and have started clinical research for IBD patients since 2010. A multicenter clinical trial is currently underway in Japan for patients with refractory ulcerative colitis and Crohn’s disease. Moreover, since current AM administration is limited to continuous intravenous infusion, the development of a subcutaneous formulation using long-acting AM is underway for outpatient treatment.

## 1. Introduction

Ulcerative colitis (UC) and Crohn’s disease (CD) are the most common types of chronic inflammatory bowel disease (IBD). The number of people with IBD worldwide reached over 6.8 million in 2017 [1]. The cause of IBD is unknown but is thought to be caused by abnormalities in intestinal immunity, a genetic predisposition, and environmental factors, such as diet and intestinal microflora.

Although there is no cure for IBD, there have been remarkable therapeutic developments that contribute to the induction and maintenance of remission over the past 20 years, including anti-TNF-alpha antibodies, tacrolimus, anti-IL-12/23p40 antibodies, Janus kinase (JAK) inhibitors, and integrin inhibitors. While these drugs have shown excellent efficacy, they also negatively affect patients’ immune systems, leading some patients to develop infections and malignant lymphomas. 

Severe infections such as tuberculosis and Pneumocystis pneumonia require serious attention, particularly in elderly patients and patients with underlying diseases, such as diabetes mellitus. Further, the above treatments demonstrate a diminishing therapeutic response, as they are immunogenic and targets of anti-drug antibodies. Therefore, the development of new therapeutic agents for IBD requires efficacy, safety, and immunogenicity.

Adrenomedullin (AM), an endogenous vasodilatory peptide, was isolated from human pheochromocytoma by Kitamura et al. in 1993 [2]. AM was initially studied as a circulatory agonist but was later found to promote angiogenesis, organ protection, and anti-inflammatory immune activity. AM is also widely expressed in the gastrointestinal tract epithelia and effectively treats gastric ulcers and enteritis in animal models. Therefore, we began translational research on the clinical application of AM in IBD [3]. 

Since AM is an endogenous bioactive peptide, it has low immunogenicity, is considered relatively safe, and is expected to be developed into a novel IBD treatment. In this article, we review the basics of AM research findings, including its effects on enteritis in animal models, and present the current status and potential of clinical research findings for IBD patients.

## 2. Structure and Biosynthesis of Adrenomedullin

AM is composed of 52 amino acids [2] and has a ring structure consisting of six amino acids and a C-terminal amide structure (Figure 1). These two structural features are essential for their biological activity. It also shares homology with calcitonin gene-related peptide (CGRP), amylin, and adrenomedullin 2/intermedin (AM2/IMD) to form the calcitonin peptide superfamily. 

AM is widely expressed throughout the blood vessels, heart, lungs, kidneys, and gastrointestinal tract and is highly concentrated in the adrenal medulla. The pro-adrenomedullin N-terminal 20 peptide (PAMP) has a shorter duration of antihypertensive activity than AM and cooperatively regulates blood circulation with AM (Figure 2) [4]. Although mid regional pro-adrenomedullin (MR-pro ADM) has no biological activity, it has been attracting attention as a biomarker for the prognosis of heart failure [5], myocardial infarction [5], community-acquired pneumonia [6], septic shock [7], and COVID-19 [8,9].

## 3. Adrenomedullin Receptors

The AM receptor consists of a complex of calcitonin receptor-like receptors (CRLR; a seven-transmembrane G protein receptor) and a receptor activity-modifying protein (RAMP; a single transmembrane receptor). There are three subtypes of RAMPs, RAMP1, RAMP2, and RAMP3, and the CRLR/RAMP1 complex has a high affinity for CGRP, while the CRLR/RAMP2 and CRLR/RAMP3 complexes have high affinity for AM. The CRLR/RAMP2 complex comprises the AM1 receptor, and the CRLR/RAMP3 complex constitutes the AM2 receptor (Figure 3).

AM contributes to the development and homeostasis of blood vessels and lymphatic vessels; AM knockout mice develop defective embryos due to defective blood and lymphatic vessel formation. Furthermore, studies using RAMP2 and RAMP3 gene knockout mice report that the AM1 receptor is involved in angiogenesis and vascular homeostasis, while the AM2 receptor regulates lymph vessel function [10]. Thus, we speculate that AM influences vascular and lymphatic system ecological functions through two receptors.

## 4. General Physiological Effects of Adrenomedullin

AM circulates the blood constitutively and is detected in healthy individuals; its expression is enhanced by mechanical stimuli, such as: Myocardial and vascular wall stretchingOrgan ischemia and hypoxiaInflammatory factors, such as inflammatory cytokines, angiotensin II, oxidative stress, and various tissue stress.

AM promotes various pathophysiological effects, such as (Table 1) [11]:VasodilationAngiogenesisCardioprotectionNephroprotectionAnti-oxidationAnti-apoptosisTissue repair and regeneration

In a study of blood levels in patients with diseases, increases in blood (AM) were observed in cardiovascular diseases such as essential hypertension, heart failure, and renal failure [12]. Elevated blood (AM) is associated with the development of inflammatory diseases such as pancreatitis and sepsis [13] in patients with IBD [14], the severity of which increases in a concentration-dependent manner.

## 5. Anti-Inflammatory Effects of Adrenomedullin

Blood (AM) is markedly elevated during severe inflammation, such as burns, pancreatitis, and systemic inflammatory response syndrome (SIRS) associated with sepsis [12].

In vitro studies using cultured vascular smooth muscle cells (VSMCs) and endothelial cells have shown that pro-inflammatory cytokines such as IL-1, tumor necrosis factor (TNF)-α, and lipopolysaccharide (LPS) stimulate AM expression in smooth muscle and endothelial cells [15]. In addition, the monocyte/macrophage cell line (RAW 264.7), murine celiac macrophages, and peripheral blood-derived monocytes have shown that monocyte-macrophage differentiation enhances AM production. Further, AM inhibits monocyte and macrophage TNF-α and IL-6 secretion following LPS-stimulation [16]. In an in vivo mouse study of sepsis treated with LPS and D-galactosamine, AM-overexpressing mice had demonstrated milder liver failure than wild-type mice, demonstrating that endogenous AM protects against SIRS [17]. Further, septic mice models treated with AM demonstrated improved hemodynamics and decreased pro-inflammatory cytokines such as TNF-α, IL-1β, and IL-6 [18] than control mice. Thus, AM ameliorates inflammatory conditions and is expected to be applied clinically to treat various inflammatory diseases.

## 6. Physiological Functions of Adrenomedullin in the Gastrointestinal Tract

AM is widely expressed throughout the mucosal epithelium, glandular duct cells, neuroendocrine cells, and smooth muscle cells of the gastrointestinal tract, between the oral cavity and the large intestine [19]. The physiological effects of AM include suppressed gastric acid secretion via somatostatin in the stomach, enhanced electrolyte secretion in the colon, suppressed gastrointestinal motility, and changes to microcirculation flux. Additionally, AM has similar physiological and antibacterial effects as defensins and may contribute to the mucosal defense system by regulating the oral and intestinal microbiome [20].

## 7. Pathophysiological Function of Adrenomedullin in Inflammatory Diseases of the Gastrointestinal Tract

The pathophysiological function of AM in gastrointestinal diseases has been reported in many studies of stomach, small and large intestine mucosal injury.

### 7.1. Effect of AM on Gastric Mucosal Injury

AM is strongly expressed in the tissues surrounding ulcers in patients with gastric ulcers, and the expression of AM tends to increase with ulcer healing, suggesting that AM may be involved in mucosal regeneration [21]. Animal models of gastric mucosal injury have demonstrated that AM ameliorates mucosal injury. It has been reported that AM inhibits gastric acid secretion [22], maintains mucosal blood flow by inhibiting gastric artery contraction [23], and promotes proliferation [24] and restructuring of mucosal epithelial cells [25]. However, specific mechanisms are under active investigation.

### 7.2. Effect of AM on Enteritis

Since AM inhibits systemic inflammation and protects the gastric mucosa, there have been many studies on the effects of AM on animal models of intestinal inflammation. We administered AM enterically to acetic acid-induced colonic ulcer model rats. We found that AM dose-dependently ameliorated colitis [26], which was in mice with dextran sulfate sodium (DSS)-induced colitis, which is a general model for IBD research [27]. 

Several research groups in Europe and Japan have reported that AM ameliorates trinitrobenzene sulfonic acid (TNBS) and DSS-induced colitis in various animal models [28,29]. Further, TNBS enteritis is more severe in AM knockout mice than in control mice [30]. Administering AM antagonists to indomethacin-induced small intestinal ulcer rat models increased the severity of enteritis [31], suggesting that endogenous, physiological AM may also relieve enteritis.

AM is reported to ameliorate intestinal inflammation by regulating mucosal immune responses, enhancing vascular, lymphatic, regenerative, barrier, and antimicrobial functions (Figure 4) [32].

#### 7.2.1. Effects on Intestinal Immunity

Animal enteritis models have shown that AM administration decreases the production of the pro-inflammatory cytokines; TNF-α, IL-6, and IFN-γ, and increases the production of the immunosuppressive cytokines; IL-10 and TGF-β in intestinal mucosa and mesenteric lymph node mononuclear cells [27,28,29,30,31].

We found that AM inhibited STAT3 and STAT1 phosphorylation in colonic epithelial cells in mice with DSS-induced colitis. Kinoshita et al. recently showed that AM inhibits NFκBp65, and STAT3 phosphorylation, suggesting AM regulates cytokine activity via the STAT3-NFκB pathway [33]. In addition, MacManus et al. reported that AM exerts intestinal epithelial anti-inflammatory effects by fine-tuning and stabilizing hypoxia-inducible factor (HIF) activity in Caco2 cells [34].

#### 7.2.2. Actions on the Intestinal Vasculature

AM contributes to the development and homeostasis of blood vessels and lymphatic vessels. Talero et al. reported that AM influences intestinal vascular function by regulating the activity of COX2 and iNOS-derived vasoactive substances in the small mesenteric arteries of TNBS enteritis rats [29]. Recently, Davis et al. generated a mouse model with a lymphatic-specific knockout of the CRLR gene (*CALCRL*) of the AM receptor and found that the mice developed intestinal lymphangiectasia. Further, administration of indomethacin prolonged intestinal inflammation [35], suggesting that AM may ameliorate intestinal inflammation through its effects on lymphatic function.

#### 7.2.3. Repair of Intestinal Epithelium and Restoration of Barrier Function

We reported that AM treatment preserved tight and adherence junctions [27], increased Klf4 expression [33], ameliorated goblet cell loss, and preserved intestinal tissue integrity in a mouse DSS enteritis model. Hayashi et al. reported that AM promoted the regeneration of injured epithelial cells via cell proliferation and wound healing assays using Caco-2 cells [36]. In addition, Yi et al. showed that AM restores intestinal epithelial barrier dysfunction by regulating myosin light chain phosphorylation [37], which is a major agonist of intestinal barrier dysfunction [38].

#### 7.2.4. Antimicrobial Effects of AM

Abnormalities in the function of the entire bacterial flora associated with dysbiosis in the intestinal microbiota have been reported in IBD patients [39]. It has been reported that Th17 cells are induced when there is an increased representation of *Proteobacteria*, including adherent/invasive *E. coli* (AIEC), in the gut microflora. Th17 cell hyperactivity is involved in the pathogenesis of IBD [40,41], which may be compounded by an accumulation of regulatory T cells with functional abnormalities, due to a decrease in butyric acid producing bacteria [42]. 

Marutsuka et al. reported that AM was distributed throughout the colonic mucosal surface layer and demonstrated dose-dependent antibacterial activity against *E. coli* [43]. Furthermore, AM administration to DSS colitis mice significantly reduced the number of anaerobic bacteria in the colon than control mice [3]. These results suggest that the antimicrobial activity of AM may mechanistically contribute to the improvement of colitis. AM treatment can restore function to dysfunctional epithelial cells and treat intestinal dysbiosis. The relationship between the antimicrobial activity of AM and the improvement of colitis requires further investigation.

## 8. Translational Research with Adrenomedullin

Since its discovery, AM has been studied as a cardiovascular agonist. Clinical trials for AM therapeutics have been initiated in Japan for patients with acute myocardial infarction, pulmonary hypertension, and peripheral arterial occlusive diseases [44]. Based on the findings of these studies, we initiated clinical trials for the development of a therapeutic drug for IBD in 2010. 

We conducted a preliminary single-center clinical trial of patients with refractory IBD. Intravenous infusion of AM 9 ng/kg/min, 8 h/day, for 14 days to patients with steroid-refractory UC resulted in 71% (*n* = 5/7) remission at 12 weeks. We observed a decrease in serum IL-6 and TNF-α 8 h after AM administration (Figure 5, case A–C) [45]. In patients with refractory Crohn’s disease with secondary failure of infliximab, continuous intravenous infusion of AM resulted in marked improvement in clinical symptoms and mucosal healing of longitudinal ulcers of the colon, as well as a re-elevation of IFX levels in the blood (Figure 5, case D) [46].

Wide and deep ulcers were observed in the colon before the AM therapy. Significant mucosal regeneration and reddening with neovascularization were observed after AM therapy (middle figures). Eight to 12 weeks after treatment with AM, the ulcers had disappeared, and ulcer scars were observed (each right figure) (modified and redrawn from [32,45,46]). UC, ulcerative colitis; CD, Crohn’s disease.

We positioned these preliminary study results as proof of concept, developed a GMP-compliant investigational new drug in 2015, and began investigator-initiated clinical trials in 2016. In a phase I clinical trial in healthy adults, we performed a single, 12-h 3, 9, and 15 ng/kg/mL AM dose study (24 patients) and a seven-day repeated-dose study (12 patients) with a placebo as the control condition. The plasma AM concentration increased in a dose-dependent manner, reached C_max_ at the end of treatment, and rapidly returned to baseline after AM administration, with a T1/2 < 60 min [47]. We confirmed that continuous intravenous AM infusion was safe and tolerable for patients in this trial. 

We conducted two multicenter RCTs in Japan: “Effects of adrenomedullin in steroid-resistant patients with ulcerative colitis: randomized, double-blind, placebo-controlled phase 2a clinical trial” and“Effects of adrenomedullin in biologics-resistant patients with Crohn’s disease: randomized, double-blind, placebo-controlled phase 2a clinical trial”.

In the phase 2 study in UC patients, 28 patients with steroid-resistant ulcerative colitis were divided into AM15, 10, and 5 ng/kg/min and placebo. There were no differences in the clinical remission rates at week two, or the primary endpoint among the four groups. However, the Mayo score at week eight was significantly lower in the high-dose AM group (15 ng/kg/min) than in the placebo group (−9.3 ± 1.2 vs. −3.0 ± 2.8, *p* = 0.035). The clinical remission rate at week eight was significantly higher than that of the placebo (3/3, 100% vs. 0/2, 0%, *p* = 0.025), and 50% of patients in the high AM dose group achieved mucosal healing, Mayo endoscopic subscore = 0. The adverse events caused by the vasodilatory effect of AM were mild, and none were serious. This study demonstrated the efficacy and safety of intravenous AM therapy in patients with steroid-resistant UC [48].

The clinical trial results for Crohn’s disease patients are currently not available for publication, and the details have been withheld.

AM therapies are expected to be developed into new IBD treatments, as its mechanism of action is not demonstrated by existing drugs, and it has a high safety profile. On the other hand, due to its short half-life of <60 min, prolonged intravenous administration was required in the aforementioned clinical trials. 

Although continuous intravenous infusion is acceptable for remission induction therapy for patients requiring hospitalization, it is not convenient for outpatient treatment or remission maintenance therapy. Therefore, we prepared polyethylene glycol-modified (PEGylated) AM for subcutaneous injection in outpatient clinics. We succeeded in extending the half-life of AM significantly. Further, the rapid decrease in blood pressure, which is a side effect of native AM administration, was suppressed, enabling the development of a hemodynamically safe treatment [49].

Furthermore, Nagata et al. created PEGylated AM with a larger molecular weight of 60 kDa and a longer half-life. They confirmed that 60 kDa PEGylated AM has the physiological and therapeutic functions of native AM in animal enteritis models (Figure 6 and Figure 7) [50]. Soon, we plan to initiate a phase I study using 60 kDa PEGylated AM. In addition, we have successfully developed IgG Fc region fusion AM [51], and human albumin-modified AM [52] and are currently conducting preclinical studies using these modified AMs.

## 9. Future Perspectives on the Development of Adrenomedullin Therapies for IBD

As described above, the safety and efficacy of AM therapy was confirmed in a relatively small number of UC patients. We are now considering the following strategies for future development.

(1)Phase 3 trials should be conducted to evaluate the safety and efficacy of AM therapy in patients with IBD refractory to steroids and biologics.(2)The efficacy and safety of remission maintenance by the intermittent administration of native AM and PEGylated AM in patients who have achieved remission with AM therapy should be evaluated (Phase 1–2).(3)The safety of long-term repeated administration of AM should be evaluated.

## 10. Conclusions

In this review, we reviewed the biosynthesis, receptor characteristics, and physiopathological effects of AM and reported on the current status of drug discovery research for novel IBD therapies. In particular, results of clinical research on AM therapy have demonstrated that a significant efficacy can be achieved at relatively low doses, where the side effect of hypotension caused by AM is not a clinical problem.

Although further validation of effective doses, dosages, and administration methods of AM therapy for IBD patients is needed, AM is considered to be a promising drug for IBD, acting through a novel mechanism.

## Figures and Tables

**Figure 1 biomedicines-09-01068-f001:**
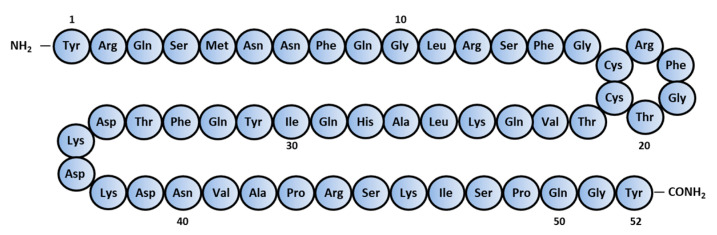
Amino-acid sequence of human Adrenomedullin.

**Figure 2 biomedicines-09-01068-f002:**
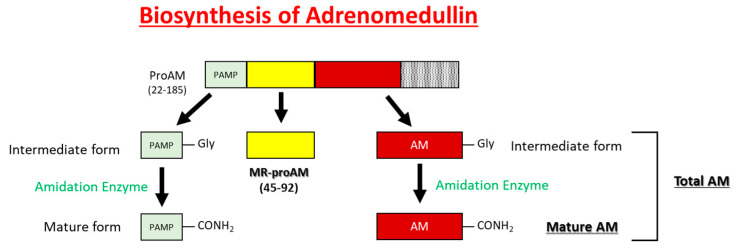
Schematic representations of the processing of AM, MR-proAM, and PAMP from proAM.

**Figure 3 biomedicines-09-01068-f003:**
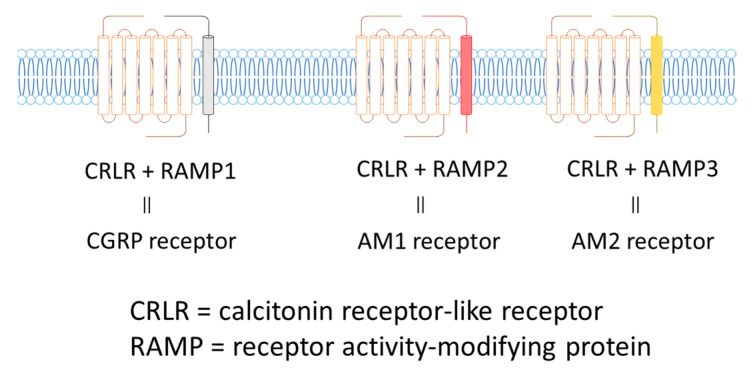
CGRP and AM Receptors.

**Figure 4 biomedicines-09-01068-f004:**
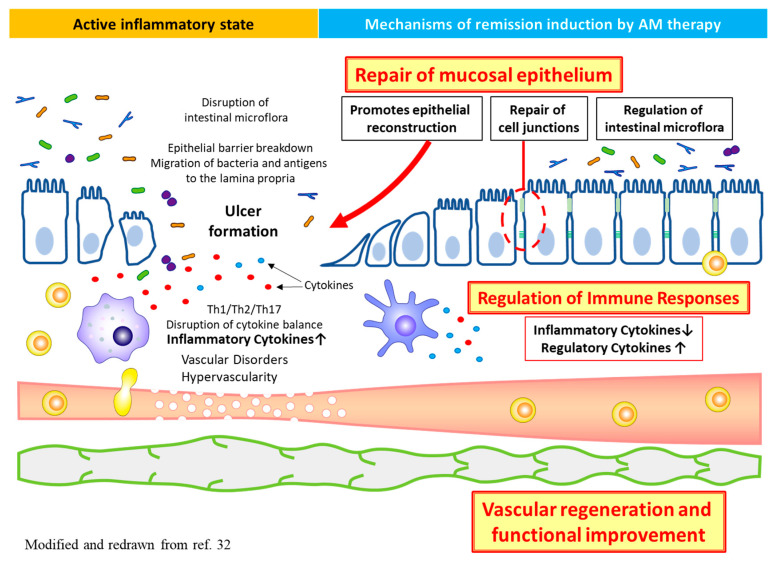
Pathophysiology of IBD and therapeutic targets of AM therapy.

**Figure 5 biomedicines-09-01068-f005:**
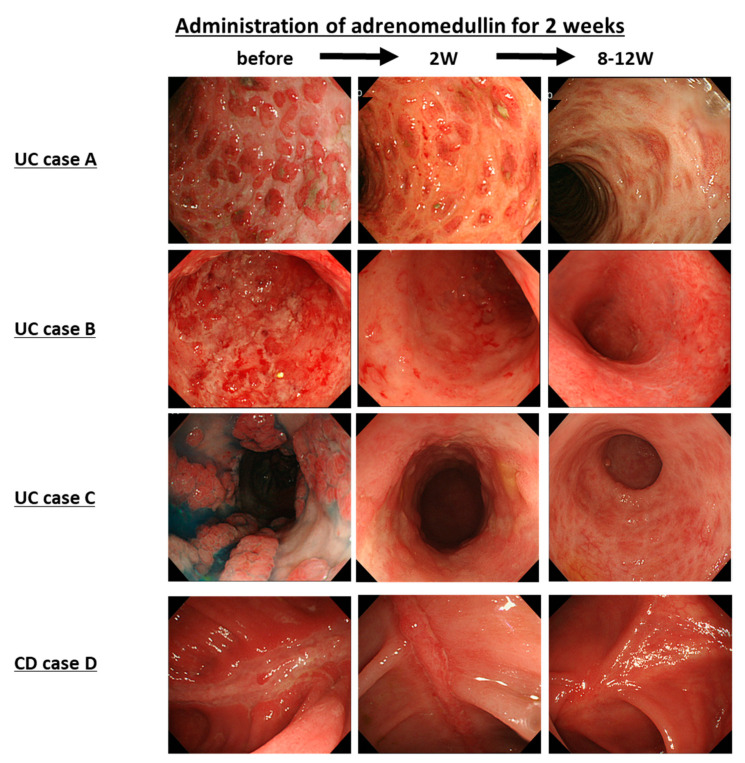
Improvement of colonoscopic findings in adrenomedullin therapy.

**Figure 6 biomedicines-09-01068-f006:**
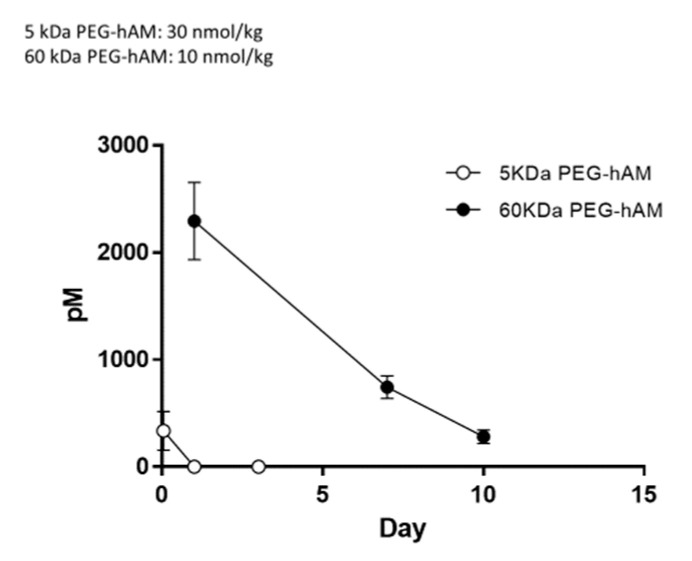
PEGylated human adrenomedullin (PEG-hAM) has a prolonged half-life in the blood. Plasma disappearance of 5 kDa PEG-hAM (open circles) and 60 kDa PEG-hAM (closed circles) over time. Five kDa PEG-AM (30 nmol/kg) and 60 kDa PEG-AM (10 nmol/kg) were injected subcutaneously into anesthetized rats, and blood samples were taken at the indicated time points (modified and redrawn from [50]).

**Figure 7 biomedicines-09-01068-f007:**
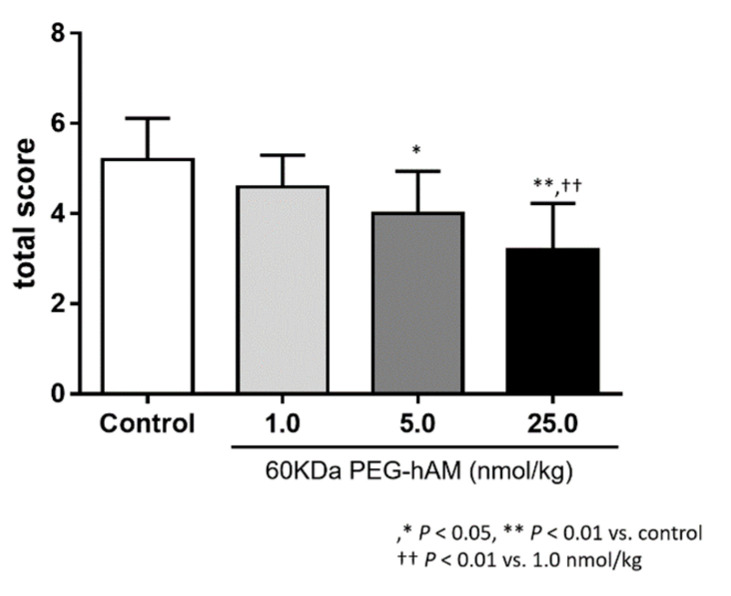
Anti-Inflammatory Effects of 60 kDa PEG-hAM on DSS-Induced Colitis. A comparison of the total inflammation score in the DSS-induced colitis model after treatment with 60 kDa PEG-hAM (*n* = 10). In mice with DSS-induced colitis, treatment with 60 kDa PEG-hAM reduced bloody stool and body weight reduction scores. The total inflammation scores were lower in the group treated with 60 kDa PEG-hAM than in controls. Results are shown as the mean ± SE. * *p* < 0.05, ** *p* < 0.01 versus control and ^††^
*p* < 0.01 versus 1.0 nmol/kg 60 kDa PEG-hAM (modified and drawn from [50]).

**Table 1 biomedicines-09-01068-t001:** Diverse physiological effects of adrenomedullin.

Hemodynamic Improvement
VasodilationDiuresisInotropic effectImprovement of intestinal blood flow
**Organ protection**
Suppression of oxidative stressInhibition of cardiac hypertrophy and fibrosisInhibition of vascular smooth muscle proliferationResistance to ischemia in heart, kidney and brain
**Anti-inflammatory effect**
Inhibition of pro-inflammatory cytokinesAntibacterial effectOrgan protection in SIRSImprovement of gastrointestinal mucosal injury
**Tissue repair and regeneration**
Vascular regeneration and stabilizationInhibition of apoptosisPromotes intestinal epithelial growth and remodelingRepair of epithelial junction molecules

## Data Availability

All the data provided in the review article.

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
