# Peer review of "Adrenomedullin: A Novel Therapeutic for the Treatment of Inflammatory Bowel Disease"

_biomedicines, 2021, doi:10.3390/biomedicines9081068_

Round 1
Reviewer 1 Report
Yours is an overall good study. The analysis of the effects of the adrenomedullin and the potential use is thorough and the scientific value is also good, especially for the refractory to steroids IBD. Although I am not sure about the adverse effects, considered as mild. Your conclusion is not confirming the use of adrenomedullin as the title suggests. The plan of the study is also good as the RCTs.
Author Response
We would like to thank you for giving us the opportunity to submit a revised version of our manuscript and the reviewers for their constructive comments that have helped to substantially improve our manuscript. We have replied to these comments in detail and revised the manuscript accordingly.
We simplified the conclusion as suggested and added statement " significant efficacy can be achieved at relatively low doses, where the side effect of hypotension caused by AM is not a clinical problem." Also checked the English language.
Reviewer 2 Report
Authors present very interesting review paper about the bioactive peptide adrenomedullin (AM). In the paper they discuss the biosynthesis, receptor characteristics, and physiopathological effects of AM. The author underline the potential effects of AM in IBD patients.
I suggest to add the short paragraph about the future perspectives of AM among IBD - what further studies are warranted to confirm the role of AM in management of IBD patients to underline the role of AM in the therapy of IBD.
Author Response
We would like to thank you for giving us the opportunity to submit a revised version of our manuscript and the reviewers for their constructive comments that have helped to substantially improve our manuscript. We have replied to these comments in detail and revised the manuscript accordingly.
We added novel section: 9. Future perspectives.... Also checked the English language.